# The use of newborn foot length to identify low birth weight and preterm babies in Papua New Guinea: A diagnostic accuracy study

**Alice Mengi**[1], **Lisa M. Vallely**[2,3], **Moses Laman**[4], **Eunice Jally**[1], **Janeth Kulimbao**[1], **Sharon Warel**[1], **Regina Enman**[1], **Jimmy Aipit**[5], **Nicola Low**[6]*, **Michaela A. Riddell**[1,3]

**1** Department of Infection and Immunity, Papua New Guinea Institute of Medical Research, Madang, Papua New Guinea, **2** Papua New Guinea Institute of Medical Research, Sexual and Reproductive Health Unit, Goroka, Papua New Guinea, **3** The Kirby Institute, Global Health Program, University of New South Wales, Sydney, Australia, **4** Department of Vector Borne Diseases, Papua New Guinea Institute of Medical Research, Madang, Papua New Guinea, **5** Department of Paediatrics, Madang Provincial Health Authority, Madang, Papua New Guinea, **6** Institute of Social and Preventive Medicine, University of Bern, Bern, Switzerland

* nicola.low@unibe.ch

**Data Availability Statement:** Data to reproduce tables and figures are available in S1 Table and S2 Table, or as a link to the Neofoot project in the Open Science Framework (osf.io).

## Abstract

Low birth weight (LBW, <2.50 kg) and preterm birth (PTB, <37 completed weeks of gestation) are important contributors to neonatal death. Newborn foot length has been reported to identify LBW and PTB babies. The objectives of this study were to determine the diagnostic accuracy of foot length to identify LBW and PTB and to compare foot length measurements of a researcher with those of trained volunteers in Papua New Guinea. Newborn babies were enrolled prospectively with written informed consent from their mothers, who were participating in a clinical trial in Madang Province. The reference standards were birth weight, measured by electronic scales and gestational age at birth, based on ultrasound scan and last menstrual period at the first antenatal visit. Newborn foot length was measured within 72 hours of birth with a firm plastic ruler. Optimal foot length cut-off values for LBW and PTB were derived from receiver operating characteristic curve analysis. Bland-Altman analysis was used to assess inter-observer agreement. From 12 October 2019 to 6 January 2021, we enrolled 342 newborns (80% of those eligible); 21.1% (72/342) were LBW and 7.3% (25/342) were PTB. The area under the curve for LBW was 87.0% (95% confidence intervals 82.8–90.2) and for PTB 85.6% (81.5–89.2). The optimal foot length cut-off was <7.7 cm for both LBW (sensitivity 84.7%, 74.7–91.2, specificity 69.6%, 63.9–74.8) and PTB (sensitivity 88.0% (70.0–95.8), specificity 61.8% (56.4–67.0). In 123 babies with paired measurements, the mean difference between the researcher and volunteer measurements was 0.07 cm (95% limits of agreement -0.55 to +0.70) and 7.3% (9/123) of the pairs were outside the 95% limits of agreement. When birth at a health facility is not possible, foot length measurement can identify LBW and PTB in newborns but needs appropriate training for community volunteers and evaluation of its impact on healthcare outcomes.

**Funding:** This study received funding from the Swiss Programme for Research on Global Issues for Development (Swiss National Science Foundation grant number IZ07Z0- 160909 to NL, ML); Joint Global Health Trials programme (United Kingdom Department For International Development/United Kingdom Medical Research Council/Wellcome Trust grant number MRN006089/1 to NL), and Australian National Health and Medical Research Council (grant number 1084429 to LMV). The funders had no role in study design, data collection and analysis, decision to publish, or preparation of the manuscript.

**Competing interests:** The authors have declared that no competing interests exist.

## Introduction

Globally, complications of preterm birth (PTB), including low birth weight (LBW), are the leading cause of neonatal death in the first week of life [1]. Identifying the most vulnerable neonates so that they can receive timely essential care is challenging in resource-limited settings, particularly in rural areas or when births occur in the community [2]. Babies born LBW (<2.50 kg) [3] are either preterm (<37 completed weeks gestation) or small for gestational age (birth weight below the 10th percentile of the distribution for gestational age and sex) [3,4]. Many babies are, however, not weighed at birth [5] or their mothers have not had an adequate assessment during pregnancy to allow gestational age at birth to be calculated [2]. These assessments require reliable equipment and skilled, trained health workers to conduct ultrasound scans during pregnancy, measure birth weight accurately, or apply clinical assessments, such as Dubowitz or New Ballard scores [6].

Foot length is one of a range of anthropometric measures that have been used as proxies for LBW and PTB in resource-limited settings [6–18]. In a systematic review and meta-analysis of studies published between 2007 and 2020, Folger et al. found 19 studies, all conducted in south and south-east Asia and sub-Saharan Africa [19]. The authors found that a newborn foot length of <7.7cm had reasonable accuracy for identification of LBW babies in 3 studies in Asia (pooled sensitivity 87.6%, 95% confidence interval, CI 55.7–97.5%, specificity 70.9%, 23.5–95.1%). They noted, however, that few studies used an accurate reference method for assessing gestational age and did not conduct a meta-analysis of foot length and PTB because of heterogeneity in study methods [19]. The optimal cut-off for newborn foot length may differ by country and region and performance characteristics could also vary, according to whether measurements are taken by clinically trained staff or community volunteers, field workers and care givers [11,20].

Papua New Guinea (PNG) is an island state in the South Pacific, which has one of the highest neonatal mortality rates in the world, with 22 newborn deaths per 1000 live births in 2020 [21,22]. Although supervised delivery at a health facility is encouraged, uptake of antenatal care in PNG is poor and high-risk babies cannot be easily identified and referred [21,23]. From 2016–2020, only around half of pregnant women in PNG had at least one antenatal visit and just over a third gave birth in a health facility. In Madang Province, about a third of pregnant women had at least one antenatal visit and about a quarter gave birth in a health facility [21]. The objectives of this study were to 1) determine the diagnostic accuracy of newborn foot length measurement to identify LBW or PTB babies in health facilities and community settings and 2) to compare foot length measurements taken by trained volunteers with those of a researcher to assess the feasibility of newborn foot length measurement in a community setting.

## Materials and methods

The Neofoot (neonatal foot length) study was a prospective single arm diagnostic test accuracy study (S1 Protocol). We report the study using the Standards of Reporting of Diagnostic accuracy 2015 (S1 Checklist). The study was conducted in three primary care clinics in Madang Province, PNG, which serve a total population of approximately 35,000, according to the 2011 PNG National Census [24]. The study population included both people living in urban shanty settlements and rural subsistence farmers in areas with poor road conditions and difficult access to health facilities. The Neofoot study was designed as a sub-study of the Women And Newborn Trial of Antenatal Interventions and Management (WANTAIM) trial, a randomised controlled trial which assessed the effect of point-of-care testing and treatment of curable sexually transmitted infections and bacterial vaginosis during pregnancy on birth outcomes [25]. Women enrolled in the WANTAIM study were aged 16 years or older at a gestational age

below 26 completed weeks, based on the date of the first day of the last menstrual period or an ultrasound scan performed by trained midwives at the first antenatal visit [26]. Their newborn babies were eligible for inclusion if the clinical researcher (AM) or senior study staff (EJ, JK, SW, RE) assessed them within 72 hours of birth, their general condition was stable and the baby did not have abnormalities. There were no additional exclusion criteria. At the first post-natal visit, the researcher or staff member used pictorial diagrams to explain the Neofoot study in Tok-Pisin to consecutive women whom they attended, assessed their understanding of the study and answered questions. The mother or guardian gave a signature or thumbprint as informed consent for their baby to take part (S1 Protocol, pages 18, 24). For women who were unable to read or write, an independent witness was present during the consent procedures and also signed the consent form.

### Reference standards

Birth weight was the average of two measurements taken at a 5 to 15 minute interval by trained WANTAIM clinical staff within 72 hours of birth using an electronic infant scale (Charder Cupid 1, Charder Medical, Taiwan) placed firmly on a hard surface and calibrated to zero [27]. Birth weight was before the index test on a separate clinical record form. LBW was defined as below 2.50 kg [3]. Gestational age at birth was measured in completed weeks of pregnancy, calculated from the best obstetric estimate of the due date [26], using the date of the first day of the last menstrual period or an ultrasound scan (Logiq V2 portable ultrasound, GE Healthcare) performed by trained midwives at the first antenatal visit. Gestational age at birth was calculated at the time of statistical analysis. PTB was defined as birth before 37 completed weeks of pregnancy [4].

### Index test

The index test was the length of the right foot, measured within 72 hours of birth. We calculated the average of two measurements, taken 5 to 15 minutes apart by the clinical researcher. Foot length was measured using a firm clear plastic 15cm ruler, bought from a local store and stuck to an A4-sized cardboard box, which was cut open (Fig 1). The tip of the ruler was pushed into a slit at the bottom of the box so that the zero level was at the base of the box. The researcher placed the right heel at the zero level and held the plantar surface of the foot against the ruler. Foot length at the tip of the hallux was measured to the nearest 1 mm, with a plastic setsquare perpendicular to the ruler.

### Training of volunteers

To assess the feasibility of community members or village health workers identifying at risk babies, we enrolled nine volunteers, who were involved in the WANTAIM trial and were numerate and literate. The clinical researcher gave a half-day training session at the postnatal ward of Madang Provincial Hospital. The volunteers learned how to measure the baby's foot length, how to complete the case record form and practised both tasks.

### Measures to reduce the risk of bias

To reduce the risk of measurement bias in foot length, the researcher and the volunteer recorded two measurements at a 5-to-15-minute interval. To ensure that the measurements were independent, the researcher and volunteer recorded their measurements on separate pages of the case record form and could not see the other person's record. Birth weight was measured before the index test on a separate clinical record form. The clinician researcher had

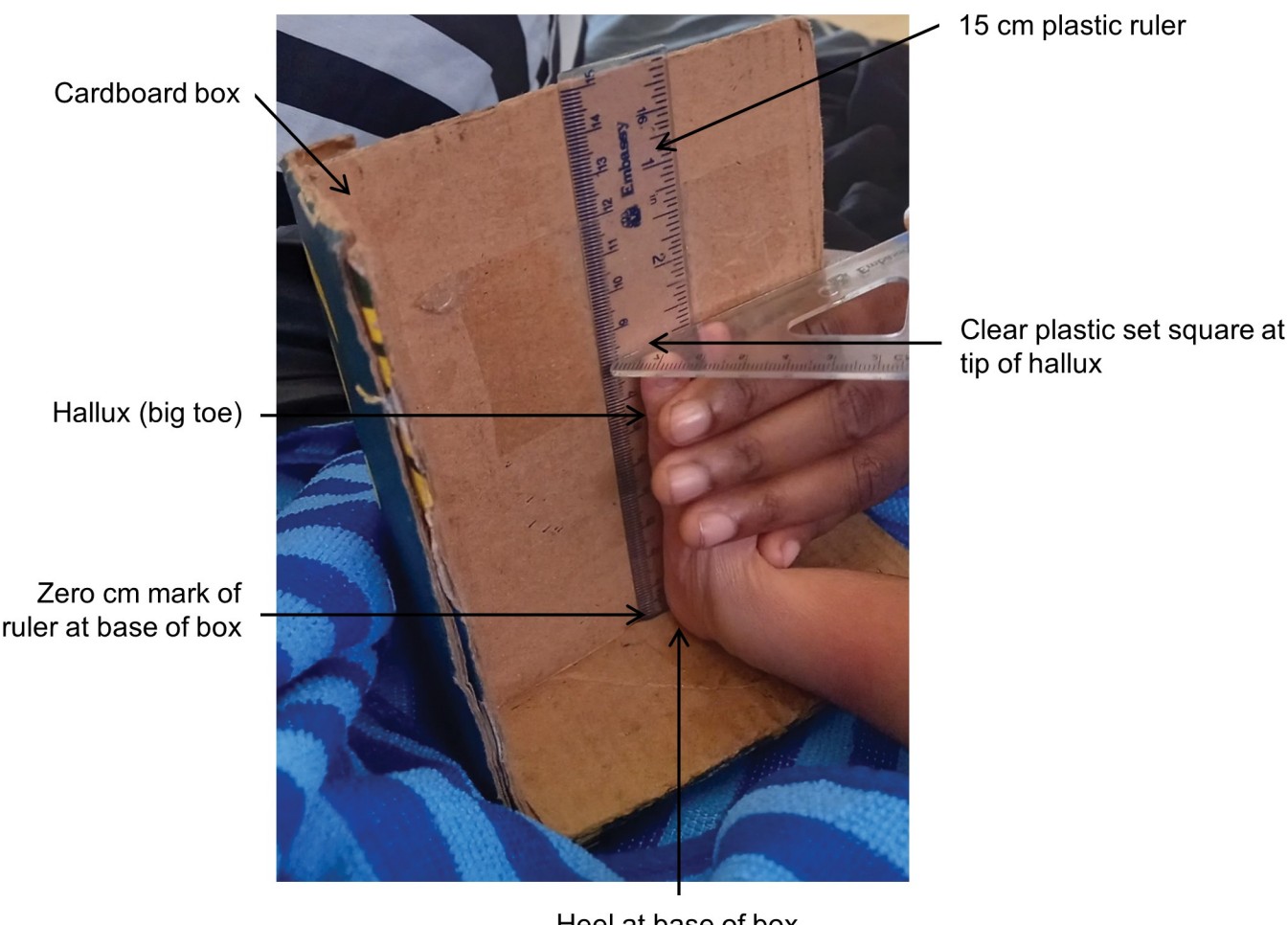

**Fig 1. Measurement of newborn foot length.** Measurements were made with a 15cm clear plastic ruler stuck to the inside of a cardboard box. The zero mark of the ruler is at the base of the box. Foot length is measured from the base of the heel to the tip of the hallux.

access to clinical information about the newborn, but the best obstetric estimate of gestational age at birth [26] was calculated at the time of statistical analysis.

## Ethics approval

The Neofoot study protocol was approved by the PNG Institute of Medical Research Institutional Review Board (IRB 1811) and the PNG Medical Research Advisory Committee (MRAC 18.18). The WANTAIM study (ISRCTN37134032) received ethical approval from the Institutional Review Board of the PNG Institute of Medical Research (IRB number 1608); the Medical Research Advisory Committee of the PNG National Department of Health (MRAC number 16.24); the Human Research Ethics Committee of the University of New South Wales (HREC number 16708); and the Research Ethics Committee of the London School of Hygiene and Tropical Medicine (REC number 12009) [25].

## Data management

Data were recorded on printed case record forms designed using TELEform™ Elite version 10.5 (https://teleform.software.informer.com/10.5/). The case record forms were checked for

discrepancies or out of range values and corrected before scanning, after which they were machine-read into a database (Microsoft). Variables collected from the mother at the first antenatal visit, as part of the WANTAIM study, included demographic details (province of birth, age, and marital status), gestational age at enrolment, estimated due date, haemoglobin, HIV test result (Determine™ HIV 1/2 Test), syphilis (SD Bioline, Abbott Diagnostics) results and use of betel nut, cigarettes, and alcohol during pregnancy. Data collected at the postnatal visit included date and place of birth, reason for giving birth at that location, newborn birth weight and sex of the baby. Data about foot length were the two measurements made by the researcher and the volunteers.

## Statistical analysis

All statistical analyses used STATA (Stata 14.2 or 16, College Station, Texas, United States). For each newborn we calculated the gestational age at birth using the estimated due date established at the enrolment visit, and the average of the two measurements of birth weight and of foot length. The distributions of continuous variables were examined using histograms and described using the mean and standard deviation (SD) or median and interquartile range (IQR). For the classification of LBW and PTB, birth weight and gestational age at birth were dichotomised in accordance with the reference standard definitions.

We constructed receiver operating characteristics curves to display sensitivity against 1 minus specificity for the classification of LBW and PTB at each 1 mm increment in foot length from the lowest to the highest value. We calculated the overall area under the curve and test performance characteristics sensitivity, specificity, positive and negative predictive values (as percentages with 95% CI). We selected the optimal foot length cut-off, separately for LBW and PTB, as the value with the best balance of sensitivity and specificity, using the foot length cut-off with the highest area under the curve. For each baby for whom paired foot length measurements were available from the researcher and the volunteer, we assessed the level of agreement using Bland-Altman analysis [28].

## Sample size calculation

The planned sample size was based on achieving an acceptable level of precision (95% CI +/- 4 to 5%) for values of sensitivity from 70 to 90%. Using projected WANTAIM trial enrolment at the study clinics, we aimed to invite around 400 women to join the Neofoot study. Assuming 95% of babies had a postnatal visit within the first 72 hours and that 90% of mothers agreed to foot length measurement, the target was to have foot length measurements for 342 newborn babies.

## Results

A total of 426 babies were born to 416 mothers enrolled in the WANTAIM trial from 12 October 2019 to 6 January 2021. Of these, 84 babies were excluded from the study for reasons outlined in Fig 2. A total of 342 babies born (80.2% of those potentially eligible) to 335 mothers were enrolled. The study population included 10 of 14 twins born to 7 mothers; 3 mothers had both twins enrolled and four mothers had one twin enrolled (the other twin died before birth or before neonatal assessment). The researcher recorded foot length measurements for all 342 babies. Owing to disruptions during the COVID-19 pandemic and staff availability, a volunteer was available to measure foot length at the time of the postnatal assessment for 123 (36.0%) babies.

Among the 335 mothers, their mean age was 25.5 (SD 5.9, range 16–44) years, 43.9% (147/ 335) were primiparous and 89.3% (299/335) were married (Table 1, S1 Data). Sixty-seven

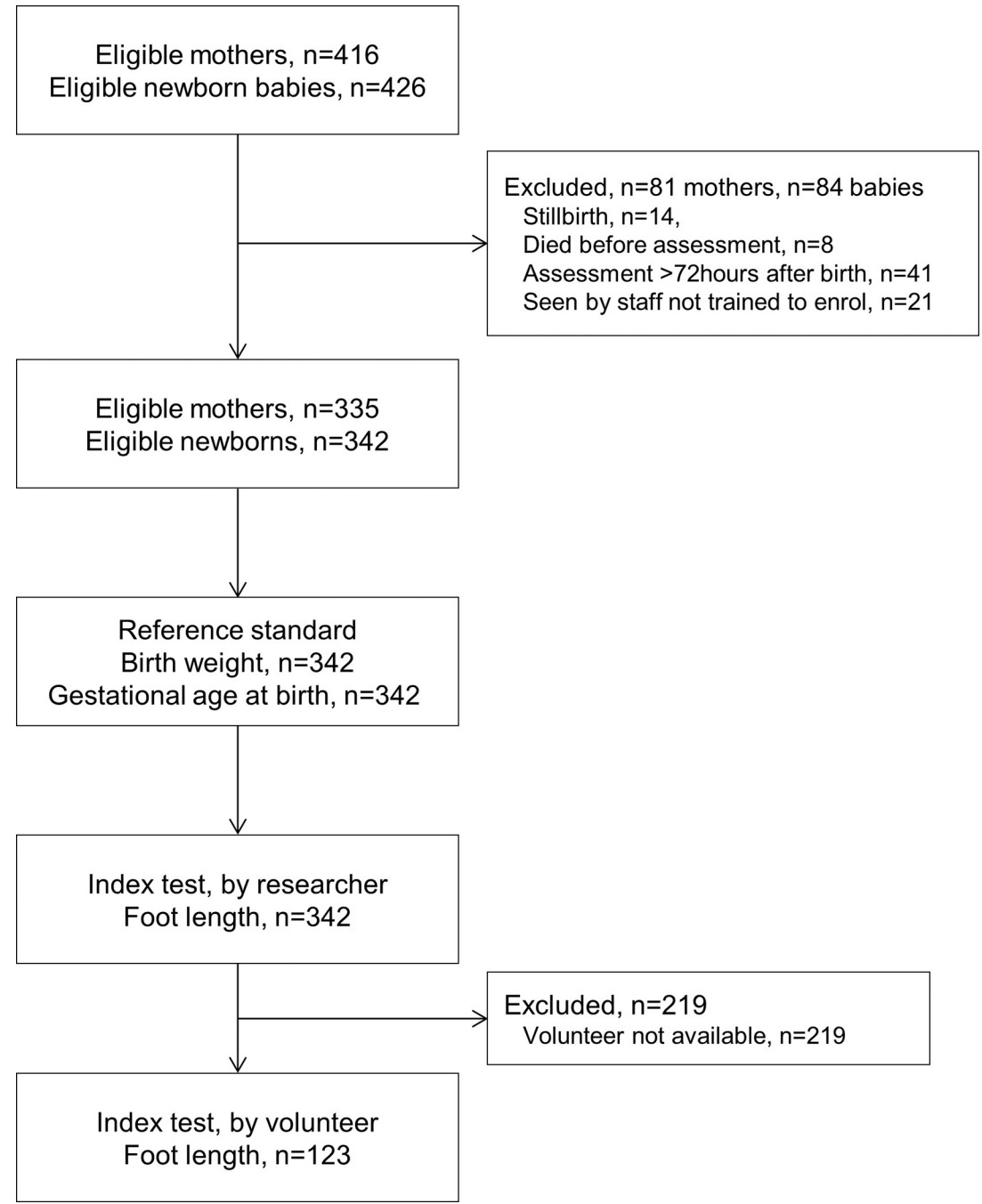

**Fig 2. Flow chart indicating number of mothers and newborn babies at each stage of the Neofoot study.**

women attended one of the primary care clinics covering urban shanty settlement areas and 268 attended clinics in rural areas. Most women reported that they currently chewed betel nut (281/335, 83.9%) and about a quarter were current smokers (81/335, 24.2%), but few (6/335, 1.8%) reported that they consumed alcohol during pregnancy. Forty-two women 12.5% (42/335) had a positive test result for syphilis and 2 women (0.6%) had a positive test result for HIV. The mean haemoglobin concentration was 8.6 g/dL (SD 1.6), nearly two thirds of

**Table 1. Characteristics of mothers whose newborn babies were enrolled in the study (N = 335).**

| Maternal characteristic | | Total N = 335 |
|---|---|---|
| Age, years | Mean (SD) | 25.5 (5.9) |
| Birthplace, region n (%) | Momase | 295 (88.1) |
| | New Guinea Island | 19 (5.7) |
| | Highlands | 16 (4.8) |
| | Southern Region | 5 (1.5) |
| Marital status, n (%) | Married | 299 (89.3) |
| | Single | 29 (8.7) |
| | Divorced/separated/widowed | 3 (0.9) |
| | Missing | 4 (1.2) |
| Chew betelnut, n (%) | Currently chew | 281 (83.9) |
| | Previously chew | 5 (1.5) |
| | Never chew | 49 (14.6) |
| Drink alcohol, n (%) | Currently drink | 6 (1.8) |
| | Previously drink | 31 (9.3) |
| | Never drink | 298 (89.0) |
| Smoke cigarettes, n (%) | Currently smoke | 81 (24.2) |
| | Previously Smoke | 34 (10.2) |
| | Never smoke | 220 (65.7) |
| Parity, n (%) | 0 | 147 (43.9) |
| | 1–3 | 141 (42.1) |
| | $\geq 4$ | 47 (14.0) |
| Current pregnancy, n (%) | Singleton | 325 (97.0) |
| | Twin | 10 (3.0) |
| Gestation in weeks, median (IQR) | USS | 22 (19.0–24.0) |
| | Fundal height | 22 (19.0–24.0) |
| | LMP | 20 (17.0–23.0) |
| HIV test result, n (%) | Positive | 2 (0.6) |
| | Negative | 323 (96.4) |
| | Not tested | 10 (3.0) |
| Syphilis test result, n (%) | Positive | 42 (12.5) |
| | Negative | 293 (87.5) |
| Haemoglobin, g/dL | Mean (SD) | 8.6 (1.6) |
| Haemoglobin, category | $\geq 10$ g/dL | 59 (17.6) |
| | 7.0–9.9 g/dL | 214 (63.9) |
| | < 7.0 g/dL | 46 (13.7) |
| | Missing | 16 (4.8) |

Abbreviations: cm, centimetres; g/dL, grams per decilitre; HIV, human immunodeficiency virus; IQR, interquartile range; LMP, last menstrual period; N, total number of participants; n, number of participants in a category; SD, standard deviation; USS, ultrasound scan.

women (63.9%; 214/335) had mild anaemia (haemoglobin 7.0–9.9 g/dL) and 13.7% (46/335) had moderate to severe anaemia (haemoglobin <7.0 g/dL).

Table 2 describes the characteristics of the 342 babies (S1 Data). Most babies (330/342, 96.5%) were born by normal vaginal birth at a health facility (72.8%; 249/342), with 27.2% (93/342) births taking place at home or before arrival at a health facility.

Birth weight was normally distributed (mean 2.8 kg, SD 0.4), with a high frequency of birth weights recorded as 2.50 kg (known as digit preference or heaping, S1 Fig).

**Table 2. Characteristics of newborn babies in the study (N = 342).**

| Newborn characteristic | | Total N = 342 | |
|---|---|---|---|
| Place of birth, n (%) | Health facility | 249 | (72.8) |
| | Born before arrival at health facility | 5 | (1.5) |
| | Home/village | 88 | (25.7) |
| Type of birth, n (%) | Normal vaginal birth | 330 | (96.5) |
| | Caesarean section | 3 | (0.9) |
| | Assisted birth | 9 | (2.6) |
| Sex, n (%) | Male | 177 | (51.8) |
| | Female | 165 | (48.3) |
| Birth weight, kg | Mean (SD) | 2.8 | (0.40) |
| | Median (IQR) | 2.8 | (2.5-3.1) |
| | Range, lowest-highest | | 1.3-4.2 |
| Birth weight category n (%) | LBW (< 2.50 kg) | 72 | (21.0) |
| | Normal birth weight (≥ 2.50 kg) | 270 | (79.0) |
| Gestational age at birth, weeks | Mean (SD) | 38.8 | (1.8) |
| | Median (IQR) | 39 | (38-40) |
| | Range, lowest-highest | | 30-42 |
| Gestational age category n (%) | Preterm < 37 weeks | 25 | (7.3) |
| | Term ≥ 37 weeks | 317 | (91.7) |
| Foot length, cm | Mean (SD) | 7.8 | (0.5) |
| | Median (IQR) | 7.8 | (7.5-8.0) |
| | Range, lowest-highest | | 5.8-8.9 |

Abbreviations: IQR, interquartile range; LBW, low birth weight; N, total number of participants; n, number of participants in a category; SD, standard deviation.

Fig 3 shows scatterplots of birth weight, gestational age at birth and the foot length measured by the researcher. About one-fifth of all babies (21.1%, 72/342), were LBW (<2.50 kg). The mean gestational at birth was 38.8 weeks (SD 1.8) and 7.3% (25/342) were preterm. Of the babies born preterm (<37 completed weeks of gestation), 5/25 babies were of normal weight (≥2.50 kg) and amongst 317 babies born at term, 52 babies had a birth weight <2.50 kg (Fig 3). Mean foot length was 7.8 cm (SD 0.5) with no adverse events recorded as a result of measurement.

## Optimal foot length cut-offs for classifying LBW and PTB

The receiver operating curves summarise the overall accuracy of foot length measurement for the classification of newborn babies as LBW or PTB (Fig 4) at cut-off increments of 1 mm. The overall area under the curve for foot length was 87.0% (95% CI, 82.3–91.6) for LBW and 85.8% (77.1–94.6) for PTB.

Optimal foot length cut-offs for the classification of LBW and PTB were determined as the value with the combination of the highest sensitivity and specificity (Tables 3, S1 and S2). The optimal foot length cut-off for identifying LBW was <7.7 cm. At this foot length, 61/72 babies were correctly classified as LBW (sensitivity 84.7%, 74.7–91.2). The birth weights of 11 babies with foot length ≥7.7 cm, who were incorrectly classified as being of normal birth weight (false negatives), ranged from 2.15–2.43 kg. Among all babies with foot length <7.7 cm, 42.7 (34.8–50.9) were LBW (positive predictive value). The negative predictive value was 94.5% (90.4–96.9).

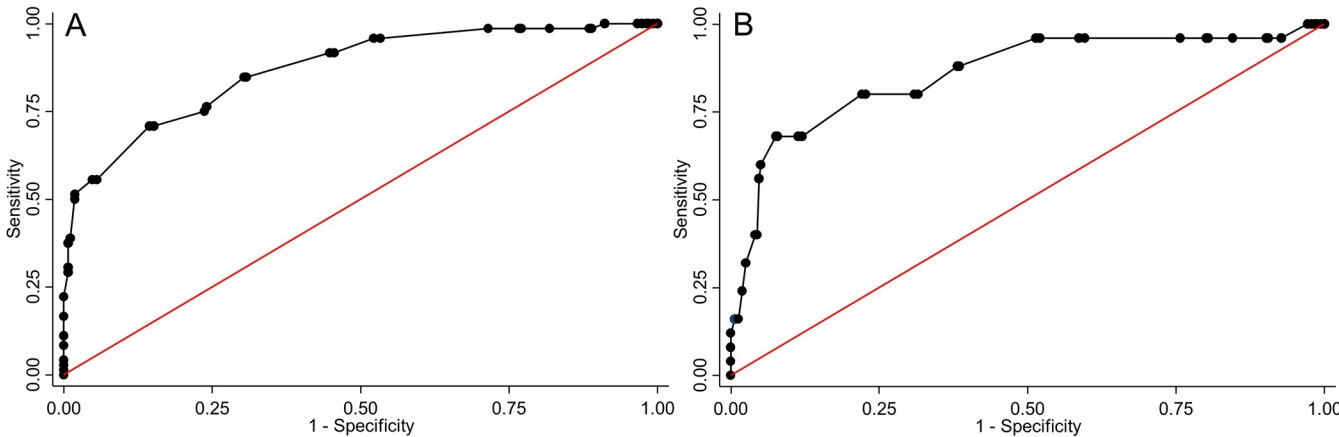

**Fig 3. Scatter plots of newborn measurements recorded within 72 hours of birth for 342 babies.** Panel A. Birth weight and foot length; B. gestational age at birth and foot length; C. gestational age at birth and birth weight. r, Pearson correlation coefficient. Reference lines in red indicate the cut-offs for low birth weight (2.50 kg) and preterm birth (37 completed weeks of gestation).

**Fig 4. Receiver operating characteristics curves of the accuracy of foot length for the classification of outcomes in 342 newborn babies.** Panel A, low birth weight, B preterm birth.

**Table 3. Diagnostic accuracy statistics at the optimal foot length cut-off for classification of babies as low birth weight or preterm.**

| Outcome | <7.7 cm | ≥ 7.7 cm | Sensitivity, % (95% CI) | Specificity, % (95% CI) | Positive predictive value, % (95% CI) | Negative predictive value, % (95% CI) |
|---|---|---|---|---|---|---|
| **Birth weight** | | | | | | |
| < 2.50 kg | 61 | 82 | 84.7 | 69.6 | 42.7 | 94.5 |
| ≥ 2.50 kg | 11 | 188 | (74.7–91.2) | (63.9–74.8) | (34.8–50.9) | (90.4–96.9) |
| **Gestational age at birth** | | | | | | |
| < 37 weeks | 22 | 121 | 88.0 | 61.8 | 15.4 | 98.5 |
| ≥ 37 weeks | 3 | 196 | (70.0–95.8) | (56.4–67.0) | (10.4–22.2) | (95.7–99.5) |

The optimal foot length cut-off for PTB was <7.7 cm (Table 3), with 22/25 babies correctly classified (sensitivity 88.0%, 70.0–95.8). Of the 3 preterm babies not identified as PTB at this cut-off (false negatives), 2 were born at 36 weeks and 1 at 35 weeks gestation; none was very (28–32 weeks) or extremely (<28 weeks) preterm. At this cut-off, 121 babies were false positives, being born at term but with foot length <7.7 cm. The positive predictive value of a foot length of <7.7 cm was 15.4% (10.4–22.2) and negative predictive value 98.5% (95.7–99.5).

Foot length measurements from both the researcher and a volunteer were available for 123 babies. Values at full or half centimetre marks were over-represented for both sets of average measurements (S2 Fig). Fig 5 shows the Bland-Altman plot, displaying the differences between the 123 paired measurements against the average of the measurements of the researcher and volunteer. The mean difference between researcher and volunteer measurements (dashed line) was 0.07 cm. The 95% limits of agreement ranged from -0.55 cm to +0.70 cm. Differences between measurements for 9/123 (7.3%) of the data pairs were outside the 95% limits of agreement.

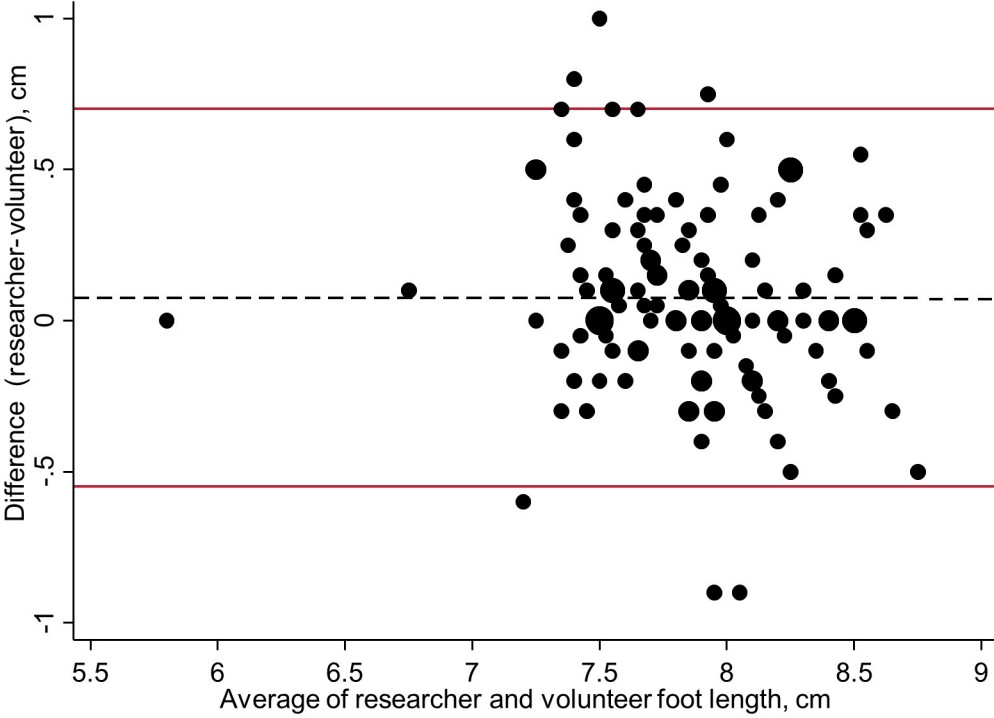

**Fig 5. Bland-Altman plot for researcher and volunteers' newborn foot length measurements (N = 123).** Size of circle is proportional to number of measurements.

## Discussion

### Summary of main findings

This prospective study enrolled 342 newborn babies in Madang Province, PNG. Overall, 21.1% (72/342) of newborns were LBW and 7.3% (25/342) were born preterm. The optimal foot length cut-off to classify LBW babies was <7.7 cm (sensitivity 84.7%, 95% CI 74.7–91.2, specificity 69.6%, 63.9–74.8) and <7.7 cm to classify PTB (sensitivity 88.0%, 70.0–95.8, specificity 61.8%, 56.4–67.0%). The 95% limits of agreement between researcher and volunteer measurements were from -0.55 to + 0.70 cm and 7.3% (9/123) of the pairs were outside these limits.

### Strengths and limitations

Strengths of this study included the use of accurate methods for determination of the reference standards for measuring birth weight and gestational age at birth. All babies were weighed twice using electronic scales [27] and gestational age at birth was calculated using published recommendations for the best obstetric estimate of the delivery date, which are based on the date of the last menstrual period and an ultrasound scan at the first antenatal visit [26]. Foot length was measured using items readily available in the community and the study involved trained volunteers from communities, with babies assessed at local health centres or at home. Also, the target sample size was reached, and outcome variables LBW and PTB were available for all babies in the study, even though a COVID-19 lock down was in place for part of the study.

There were also limitations of the study methods. First, the measurement of birth weight was taken within the first 72 hours of birth. Ideally, birth weight should be measured within a few hours of birth, before any postnatal weight loss has occurred [27]. However, Marchant et al. found that foot length increased by only 0.2 cm (SD 0.3) between the first and fifth day after birth in a study in Tanzania [10], so the delay of up to 72 hours in our study should not have affected the results. Second, the researcher and volunteers could not be blinded from the clinical status of the newborn, which might have influenced their foot length measurements. Knowledge of birthweight could have resulted in overestimation of the accuracy of classification of LBW. Since the overall areas under the curve for LBW and PTB were similar, we think risk of bias was low. Third, whilst mothers were enrolled at primary care clinics and most gave birth at a health centre or at home/village, the study was done in only one coastal province of PNG. The 2020 PNG National Department of Health Sector Review Performance found that Madang Province was one of the provinces with the highest percentage of low birth weight babies (13% in 2020) [21]. Optimal foot length cut-off values might be different in other regions of PNG or other Pacific islands.

### Comparison with other studies

Our study from PNG in the Pacific region adds to the evidence from previously published studies that have explored foot length as a proxy to identify LBW and PTB and were conducted in countries in Asia and Africa. The levels of LBW (around 20%) and PTB (around 7%) in the Neofoot study were similar to those of several other published studies that have assessed the diagnostic accuracy of foot length both in sub-Saharan Africa [10,15] and south Asia [29]. Our study in PNG, however, overcame limitations of assessment of gestational age and of applicability of findings from tertiary care setting that were identified in previous studies [19]. The Neofoot study shared several characteristics with a study by Lee et al. among 710 babies in the community in Bangladesh, but with different findings about the accuracy of foot length as a

method to identify PTB [29]. Both studies were nested in randomised controlled trials, the investigators assessed babies within 72 hours after birth, estimated gestational age based on ultrasound scan and used a clear plastic ruler to measure foot length. Lee et al. found that foot length measured at home visits by trained community health workers did not classify PTB well, with an overall area under the curve of 51.9% and sensitivity at a cut-off of <7.7 cm of 28%. In contrast, we found an overall area under the curve of 85.6% and sensitivity of 88.0% for the same cut-off. Lee et al. attributed the poor performance of foot length (and other anthropometric measurements) to the high level of foetal growth restriction (32.4% of babies were assessed as being small for gestational age). In our study, however, levels of PTB and mean birth weight were similar to those of Lee et al. and 12.3% (42/342) of term babies were LBW. One factor that might have contributed to the different findings was that all babies in Lee et al.'s study were assessed at home by community health workers, whilst more babies in our study were assessed at a health facility or community by a paediatrician researcher, where the conditions for measuring foot length might have been easier. Given the study setting, the proportion of supervised deliveries in our study population was higher than the average for Madang Province [21]. Our study could be repeated among women who do not give birth at a health facility to assess the performance of foot length measurement by a community health worker.

## Interpretation of the findings

In the Neofoot study, a foot length cut-off of <7.7 cm had similarly high overall accuracy for classification of LBW and PTB (Table 3). A common cut-off for identifying LBW and PTB was also found by Paulsen et al. (≤7.7 cm) [30] and Marchant et al. (<8 cm) in Tanzania [10]. The cut-off of <7.7 cm for LBW concurs with the pooled result from the meta-analysis of 15 studies by Folger et al., which included studies from a range of countries [19]. The lower prevalence of PTB, compared with LBW, results in a lower positive predictive value. At the optimal foot-length cut-off, around 15% of babies were preterm, according to the reference standard (compared with nearly half of babies classified as LBW). The referral of substantial numbers of babies for urgent care would increase the workload for health systems and should be assessed. The use of a plastic ruler was an appropriate technology for the measurement of foot length and taking the average of two measurements should have reduced error, although we still observed digit preference for measurements at the half or full centimetre (S2 Fig). A single training session for lay people and health facility staff to measure foot length resulted in average foot length measurements that were similar to those of the researcher, but paired measurements for individual babies differed by up to 1 cm. These findings were comparable to those of Marchant et al. who assessed 142 paired measurements, using a plastic ruler, between researchers and community volunteers in Tanzania [11]. As in our study, they found no overall bias but volunteers' measurements were shorter than those of researchers.

## Implications for research and practice

The goal of universal health coverage to ensure that all sick and small newborns survive and thrive will take time to achieve; in the meantime, there is a need for innovative ways to simplify assessment of gestational age at birth [2]. The findings of this study have implications for future research and practice in settings such as PNG, where it is difficult to identify high risk babies in populations with rural communities, high levels of home births and limited access to health facilities with skilled staff and equipment. Most women in PNG attend antenatal clinic late in pregnancy, often in the last trimester, so assessment of gestational age at birth is particularly challenging. Even though the Neofoot study was nested in a trial following Good Clinical

Practice guidelines, with ultrasound-assisted pregnancy dating and efforts to encourage supervised delivery, more than a quarter of women did not give birth at a health facility. Most of these women lived in hamlets, which are an hour or more from the main road by foot, and from which access to the labour ward at the health centre was difficult, especially at night. In settings such as these, validation of newborn footlength as a surrogate for LBW and PTB is important. Our study adds to the published evidence [19], showing that both LBW and PTB can be classified accurately and could be used by trained health workers and community health volunteers. Newborn foot length has not, however, been widely implemented. Implementation research studies should develop appropriate technologies to apply optimal cut-off measurements, training programmes that reduce misclassification from digit preference (heaping) and examine the impacts on referrals to health systems and health outcomes. In this study in PNG, newborn foot length was a simple and feasible method for the identification of LBW and PTB babies and could help to improve neonatal health outcomes.

## Supporting information

**S1 Checklist. Standards of Reporting of Diagnostic accuracy studies 2015 checklist.**
(https://www.equator-network.org/reporting-guidelines/stard/).
(DOCX)

**S1 Fig. Bar chart of birth weight, measured on electronic scales (N = 342).**
(TIF)

**S2 Fig. Bar chart of foot length measurements, average of two measurements with a plastic ruler.** Panel A, researcher (N = 342), B, volunteer (N = 123).
(TIF)

**S1 Table. Receiver operating curve statistics for accuracy of newborn foot length to classify low birth weight, 1 mm increments (N = 342).** Abbreviations: CI, confidence interval; LHR, likelihood ratio; PV, predictive value.
(DOCX)

**S2 Table. Receiver operating curve statistics for accuracy of newborn foot length to classify preterm birth, 1 mm increments (N = 342).** Abbreviations: CI, confidence interval; LHR, likelihood ratio; PV, predictive value.
(DOCX)

**S1 Protocol. Neofoot study protocol.**
(DOCX)

**S1 Data. Link to datasets.**
(DOCX)

## Acknowledgments

This study was conducted during the Master of Medical Sciences, University of Papua New Guinea of Alice Mengi. We sincerely thank all the parents of the 342 babies that participated, the PNG Institute of Medical Research and WANTAIM staff, especially Kelly Masil for scanning and verification of data and Andrew Vallely and William Pomat for acquisition of funding and oversight of the WANTAIM trial. We thank Odile Stalder at the Clinical Trials Unit, University of Bern, who helped with the statistical analyses. We thank Sister Hilda Marijembi, Sister Susie Yuangi, Naida Samson, Sister Bridegette Nawali, Lina Puli, Josephine Fei, Christophilda Noubiri and Alvita Jimmy for assistance with data collection.

## Author Contributions

**Conceptualization:** Alice Mengi, Lisa M. Vallely, Moses Laman, Nicola Low, Michaela A. Riddell.

**Data curation:** Alice Mengi, Michaela A. Riddell.

**Formal analysis:** Alice Mengi, Nicola Low, Michaela A. Riddell.

**Funding acquisition:** Lisa M. Vallely, Moses Laman, Nicola Low.

**Investigation:** Alice Mengi, Eunice Jally, Janeth Kulimbao, Sharon Warel, Regina Enman.

**Methodology:** Alice Mengi, Lisa M. Vallely, Nicola Low, Michaela A. Riddell.

**Project administration:** Michaela A. Riddell.

**Supervision:** Lisa M. Vallely, Moses Laman, Jimmy Aipit, Nicola Low, Michaela A. Riddell.

**Writing – original draft:** Alice Mengi, Lisa M. Vallely, Nicola Low, Michaela A. Riddell.

**Writing – review & editing:** Alice Mengi, Lisa M. Vallely, Moses Laman, Eunice Jally, Janeth Kulimbao, Sharon Warel, Regina Enman, Jimmy Aipit, Nicola Low, Michaela A. Riddell.

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
