## [Decision Letter · Decision Letter 0]

22 Mar 2023

PGPH-D-23-00319

The use of newborn foot length to identify low birth weight and preterm babies in Papua New Guinea: A diagnostic accuracy study

Dear Dr. Low,

Thank you for submitting your manuscript to PLOS Global Public Health. After careful consideration, we feel that it has merit but does not fully meet PLOS Global Public Health’s publication criteria as it currently stands. Therefore, we invite you to submit a revised version of the manuscript that addresses the points raised during the review process.

In addition to the comments by the two reviewers, I think it will be helpful to explain the context better. When babies are born in health facilities, presumably, actual weight can be obtained and there is no real need to resort to foot length as a proxy. It is when babies are born at home or in remote areas, then surrogates like foot length might be helpful? Surely, we want more women to be born in facilities and we want babies to be actually weighed. So, it is important to make sure that message is clearly conveyed.

In the above context, most babies in this study were born in health facilities and would have had easy access to actual weighing scales. Would it not be necessary to validate these findings in babies born at homes? Isn't that the intended use population? This limitation needs more discussion.

We look forward to receiving your revised manuscript.

Kind regards,

Madhukar Pai, MD, PhD

Editor-In-Chief

Journal Requirements:

Additional Editor Comments (if provided):

Reviewers' comments:

Reviewer's Responses to Questions

**Comments to the Author**

1. Does this manuscript meet PLOS Global Public Health’s publication criteria? Is the manuscript technically sound, and do the data support the conclusions? The manuscript must describe methodologically and ethically rigorous research with conclusions that are appropriately drawn based on the data presented.

Reviewer #1: Yes

Reviewer #2: Yes

2. Has the statistical analysis been performed appropriately and rigorously?

Reviewer #1: Yes

Reviewer #2: Yes

3. Have the authors made all data underlying the findings in their manuscript fully available (please refer to the Data Availability Statement at the start of the manuscript PDF file)?

Reviewer #1: Yes

Reviewer #2: Yes

4. Is the manuscript presented in an intelligible fashion and written in standard English?

Reviewer #1: Yes

Reviewer #2: Yes

5. Review Comments to the Author

Reviewer #1: I think the authors did a great job explaining every section of the paper. The aim/objective, methods, results section was very well presented; easy to understand. However, the authors could have written more for the discussion and explained their findings more than they have.

Reviewer #2: The authors have conducted a very important, detailed and nuanced study. The analysis is thorough, and the paper is well written.

I would advise a minor revision to adjust the discussion section subheadings to: 1) Summary of main findings 2) interpretation of the findings 3) comparison with other studies 4) strengths and limitations 5) implications for research and practice

6. PLOS authors have the option to publish the peer review history of their article (what does this mean?). If published, this will include your full peer review and any attached files.

**Do you want your identity to be public for this peer review?** For information about this choice, including consent withdrawal, please see our Privacy Policy.

Reviewer #1: **Yes: **Rishav Das

Reviewer #2: No

---

## [Editor Report · Decision Letter 1]

2 Jun 2023

The use of newborn foot length to identify low birth weight and preterm babies in Papua New Guinea: A diagnostic accuracy study

PGPH-D-23-00319R1

Dear Prof Low,

We are pleased to inform you that your manuscript 'The use of newborn foot length to identify low birth weight and preterm babies in Papua New Guinea: A diagnostic accuracy study' has been provisionally accepted for publication in PLOS Global Public Health.

Best regards,

Madhukar Pai, MD, PhD

Editor-In-Chief
